# Ameliorative Effects of *Aspergillus awamori* against the Initiation of Hepatocarcinogenesis Induced by Diethylnitrosamine in a Rat Model: Regulation of *Cyp19* and *p53* Gene Expression

**DOI:** 10.3390/antiox10060922

**Published:** 2021-06-07

**Authors:** Doaa H. Assar, Abd-Allah A. Mokhbatly, Emad W. Ghazy, Amany E. Ragab, Samah Abou Asa, Walied Abdo, Zizy I. Elbialy, Nora Elbialy Mohamed, Ali H. El-Far

**Affiliations:** 1Clinical Pathology Department, Faculty of Veterinary Medicine, Kafrelsheikh University, Kafrelsheikh 33516, Egypt; abdallah.makhbatly@vet.kfs.edu.eg (A.-A.A.M.); emad_aboamsan@vet.kfs.edu.eg (E.W.G.); nora.elbialy@gmail.com (N.E.M.); 2Pharmacognosy Department, Faculty of Pharmacy, Tanta University, Tanta 32527, Egypt; amany.ragab@pharm.tanta.edu.eg; 3Pathology Department, Faculty of Veterinary Medicine, Kafrelsheikh University, Kafrelsheikh 33516, Egypt; samahsalem@vet.kfs.edu.eg (S.A.A.); walid.eid@vet.kfs.edu.eg (W.A.); 4Department of Fish Processing and Biotechnology, Faculty of Aquatic and Fisheries Sciences, Kafrelsheikh University, Kafrelsheikh 33516, Egypt; 5Department of Biochemistry, Faculty of Veterinary Medicine, Damanhour University, Damanhour 22511, Egypt; 6Scientific Chair of Yousef Abdullatif Jameel of Prophetic Medicine Application, Faculty of Medicine, King Abdulaziz University, Jeddah 21589, Saudi Arabia

**Keywords:** diethylnitrosamine, hepatocellular carcinoma, *Aspergillus awamori*, antioxidant, UPLC-PDA-MS/MS

## Abstract

Hepatocellular carcinoma (HCC) is the most common cancer in humans. Despite advances in its treatment, liver cancer remains one of the most difficult cancers to treat. This study aimed to investigate the ameliorative action and potential mechanism of *Aspergillus awamori* (ASP) administration against the initiation process of liver carcinogenesis induced by diethylnitrosamine (DEN) in male Wistar rats. Seventy-two male rats were divided equally into eight groups as follows, Group 1: untreated control; Group 2: DEN (200 mg/kg bw) intra-peritoneally for the initiation of HCC; Groups 3–5: DEN + ASP at a dose of 1, 0.5, and 0.25 mg/kg bw and groups 6–8: ASP at a dose of 1, 0.5, and 0.25 mg/kg bw. Supplementation of *A. awamori* significantly lightened the adverse impacts induced by DEN via restoring the leukogram to normal, lowering the elevated serum aspartate aminotransferase (AST), alanine transaminase (ALT), and γ-glutamyl transferase (GGT), and alkaline phosphatase (ALP). Furthermore, it enhanced the hepatic antioxidant capacity through increasing the reduced glutathione (GSH) level and catalase (CAT) activity with a marked reduction in malondialdehyde (MDA) level. In addition, it decreased the positive GST-P foci. Likewise, a significant alteration of DEN-associated hepatocarcinogenesis occurred through inhibiting cytochrome P450 (*Cyp19*) and activating *p53* gene expression. In conclusion, supplementation of *A. awamori* counteracts the negative effects of DEN, inhibits the early development of GST-P-positive foci and could be used as a new alternative strategy for its chemo-preventive effect in liver cancer. To the best of our knowledge, the present study is the first to report the hepato-protective effect of *A. awamori* in induced hepatocarcinogenesis.

## 1. Introduction

Liver cancer is one of the world’s most prevalent diseases, especially in Asia and Africa [1]. Hepatocellular carcinoma (HCC), the fourth most common cause of cancer mortality, accounts for 90% of all liver cancer [2,3]. Hepatitis viral infection, alcohol, aflatoxins, environmental and industrial pollutants, and air and water contamination are the most common factors linked with liver cancer [4]. Diethylnitrosamine (DEN) is a well-known hepatocarcinogenic agent found in cigarette smoke, water, cured and fried foods, cheese from cheddars, chemicals from the farming industry, cosmetics, and pharmaceutical products [5,6]. By inhibiting several enzymes involved in the DNA repair process, DEN causes liver cancer in the experimental animal model [7,8] in addition to activation of cytochrome P450 enzymes, forming reactive electrophile enzymes that cause oxidation stress, which contributes to cytotoxicity, mutagens and cancers [9]. Northern blot and dot blot analysis has shown that GST-P mRNA is abundant in preneoplastic lesions and HCC induced by genotoxic carcinogens [10]. GST-P has been documented to be the most definite marker for altered hepatocellular foci in rats. Even single hepatocytes noticed to be immunoreactive to GST-P and appearing early in hepatocarcinogenesis may sometimes characterize initiated cells [11]. Cell proliferation and metabolism are essential parts of the carcinogenic process. Cell proliferation contributes to inducing the DNA mutation during the initiation of tumors [12]. Most chemical carcinogens are procarcinogens that undergo metabolic activation to be true carcinogens [13]. This bio-activation occurs in the liver where the metabolic enzymes exist as cytochrome P450 (CYP), which play crucial roles in carcinogenesis.

Despite continuous advances in managing chronic liver diseases, tumor detection and treatment, the prognosis of HCC remains lower in comparison to other tumors [14]. The incidence of high mortality and associated side effects following chemotherapy and/or radiotherapy increase the demand for alternative medicines for cancer treatment. Not surprisingly, many potent anticancer compounds have been isolated from plants, e.g., doxorubicin, taxol, etoposide, cisplatin, vinblastine, vindesine, vincristine and topotecan [15]. This has lead to the current interest in alternative medicine that will aid in therapy and help to avoid its unjustifiable use.

Currently, naturally derived compounds have notable importance for their massive application as antioxidants to inhibit free radical species and protect the cellular organelles from oxidative damage [16]. Secondary, micro-organism products have important food and nutraceutical industry applications [17]. Filamentous fungi produce numerous secondary metabolites used for antibiotic and pharmaceutical action, including anticancer and antioxidant effects [18]. Alkaloids, tannins, phenolics, steroids, and flavonoids are abundant in fungal strains [19]. Yokoyama et al. [20] identified *Aspergillus awamori*, a variant of *Aspergillus niger*, called “koji” in Japan. The genera *Aspergillus* contributes significantly to fungal origin secondary metabolites with numerous biological advantages [21]. Dietary supplementation of *A. awamori* has been reported to boost growth efficiency and reduce skeletal muscle lipid peroxidation [22,23]. Chemical analysis of *A. awamori* revealed coumarins, glucose, saponins, flavonoids, and tannin, which have antioxidant properties [24]; however, these metabolites were not fully identified.

Hence in the present study, we used the DEN-induced hepatic cancer model in Wistar rats and evaluated the cancer inhibition effects of *A. awamori* and identified the potential bioactive chemical constituents.

To the best of our knowledge, there are no previous reports on the anti-cancer effect of *A.awamori* and the underlying genetic mechanism against DEN-induced hepatic cancer in Wistar rats.

## 2. Materials and Methods

### 2.1. Ethical Statement

All procedures used in the current study were carried out based on the recommended NIH Guide for the care and use of laboratory animals by the Faculty of Veterinary Medicine Ethics Committee, Kafrelsheik University, Egypt. All precautions were followed to diminish animal suffering during the experiment.

### 2.2. Diethyelnitrosamie (DEN) Preparation

Diethylnitrosamine powder was obtained from Sigma-Aldrich Co. (St. Louis, MO, USA) and diluted in normal saline (1 gm DEN/ 25 mL saline).

### 2.3. Aspergillus awamori Extract Preparation

*A. awamori* powder (25 × 10^4^ cells per g.) was provided from Biogenkoji Research Institute, Kagoshima, Japan. *A. awamori* powder was suspended in saline with a concentration of 3 mg/6 mL and was used as a stock solution and prepared regularly by the same concentration.

### 2.4. UPLC-PDA-MS/MS for Metabolite Analysis

For metabolite analysis, *A. awamori powder* (8 g) was mixed with 80% aqueous ethanol, heated at 40 °C for 10 min, and kept overnight at room temperature. The extract was filtered and evaporated under vacuum to yield a yellowish-brown residue. UPLC was performed using a Nexera-i LC-2040 liquid chromatography system (Shimadzu, Kyoto, Japan) equipped with UPLC Shim-pack Velox C18 Column, 2.1 × 50 mm;2.7 μm particle, applying the following gradient (solvent A: water containing 0.1% formic acid; solvent B: acetonitrile) at a flow rate of 0.2 mL/min: 0–2 min: 10% B; 2–5: linear-gradient to 30% B; 5–15 min: linear-gradient to 70% B; 15–22 min: linear-gradient to 90% B; 22–25 min: linear-gradient to 95% B; 25–26 min: linear-gradient to 100% B; 26–29 min: isocratic 100% B; 29–30 min: linear-gradient to 10% B. The sample (3 mg/mL) was prepared by ultrasonication of the extract in HPLC methanol for 10 min followed by centrifugation, 3 μL of the solution was injected into the system. Compounds were detected using an LC-2030/2040 PDA detector and an LC-MS 8045 triple quadrupole mass spectrometer equipped with electrospray ionization (ESI) source in negative and positive mode (Shimadzu, Kyoto, Japan) using the following settings: nebulizer gas N_2_, 3 L/min, 4 bar; dry gas N_2_, 10 L/min, 400 °C; capillary voltage −4 kV; endplate offset −4.5 kV; collision energy 8 eV (Full MS) or 20–35 eV (MS/MS).

### 2.5. Determination of the Total Content of Flavonoids and Polyphenols

For measuring the total flavonoids content, serial dilutions of *A. awamori* extract were analyzed colorimetrically according to the procedures of aluminum chloride method using rutin as a standard [25]. The total polyphenol content was determined following the Folin–Ciocalteu method using gallic acid as a standard [26]. The contents were expressed as mg/g equivalent of the corresponding standard for each method.

### 2.6. Experimental Animals

Seventy-two male Wistar rats of average weight 120 ± 15 g were used in the present study and obtained from the Animal House Colony of the Tanta Center. Rats were acclimatized for one week to laboratory conditions (temperature 22–25 °C, relative humidity 50–60%, and 12-h photoperiods with lights on 07:00–19:00 h) in stainless steel wire-mesh cages. The animals were provided with basal diet (Appendix A) and water ad libitum throughout the experiment.

### 2.7. Experimental Protocol

After the adaptation period, the rats were randomly divided equally into eight groups (9 animals each) and treated as follows:

Group 1 Normal control untreated rats fed with standard diet and administered orally with 0.5 mL normal saline by gastric intubation.

Group 2 Rats were injected with DEN to induce hepatocarcinogenesis by providing a single dose of intraperitoneal injection (i.p) of DEN (200 mg/kg *BW*) on the first day of the experiment [27].

Groups 3, 4 and 5 Rats treated with *A. awamori* after the administration of DEN at a dose of (1, 0.5 and 0.25 mg/kg BW respectively for seven days) orally by gastric intubation (DEN + ASP 1, DEN + ASP 0.5, DEN + ASP 0.25 respectively).

Groups 6,7 and 8 Rats treated with *A. awamori* alone orally by gastric intubation daily for seven days at a dose of 1, 0.5 and 0.25 mg/kg BW respectively for seven days (ASP 1, ASP 0.5, ASP 0.25 respectively).

The experimental design is presented in Appendix A.

At the end of the experimental period, two blood samples were withdrawn from the retro-orbital venous plexus of each rat under anesthesia with intravenous injection of sodium pentobarbital (30 mg/kg BW). These samples were immediately divided as follows: one with anticoagulant for hematological examination and the other left to clot; serum was separated from clotted blood by centrifugation at 3000 rpm for 10 min for biochemical analysis. Then, the rats were sacrificed by decapitation. Livers were collected and rapidly excised from each animal, trimmed and divided into two parts; the first part was taken directly, snap-frozen in liquid nitrogen, and stored at −80 °C for evaluating lipid peroxidation, antioxidant status and real-time reverse transcription-polymerase chain reaction (RT-PCR) analysis. The second part was used for histopathological examination and immunohistochemistry.

### 2.8. Hematological Examination

Hematological parameters including red blood cell (RBC) count, hemoglobin (Hb), hematocrit (HCT), mean corpuscular volume (MCV), mean corpuscular hemoglobin concentration (MCHC), mean corpuscular hemoglobin (MCH), white blood cells (WBCs), and lymphocyte, monocyte, and granulocyte counts were estimated using Orphee Mythic 22 CT Hematology Analyzer (Orphee SA Co., Plan-les-Ouates, Switzerland).

### 2.9. Serum Biochemical Analysis

Serum alanine aminotransferase (ALT), aspartate aminotransferase (AST), alkaline phosphatase (ALP), and gamma-glutamyl transferase (GGT), total proteins (TP), and albumin were determined using the commercial kits Bio-diagnostic Co. (Giza, Egypt).

### 2.10. Hepatic Oxidative Stress and Antioxidant Status Estimation

The liver from each experimental animal was dissected out, weighed, then washed with 50 mmol/L sodium phosphate-buffered saline (100 mmol/L Na_2_HPO_4_/NaH_2_PO_4_, pH 7.4) in an ice-containing medium, with 0.1 mmol/L EDTA to remove RBCs and clots. Samples were taken and homogenized in 5–10 mL cold buffer per gram of tissue and were centrifuged at 2000× *g* for 30 min. The resulting supernatants were subjected to biochemical determination of malondialdehyde (MDA), reduced glutathione (GSH), and catalase (CAT) according to the manufacturers’ instructions (Bio-diagnostic Co., Giza, Egypt).

### 2.11. RNA Extraction and Real Time-PCR

Total ribonucleic acid (RNA) was extracted from the liver tissue of both control and treated rats according to a TRIzol reagent protocol using DNase treatment Kit (Invitrogen, Carlsbad, CA, USA) to determine the gene expression level of *Cyp19* and *P53*. First-strand cDNA was synthesized using the SuperScript III First-Strand Synthesis System for RT-PCR (Invitrogen Co., Carlsbad, CA, USA). Quantitative real-time reverse transcription-polymerase chain reaction (RT-PCR) analysis was performed using a SYBR Premix Ex Taq™ (Takara Bio Inc., Shiga, Japan). The primer sequences are listed in Table 1 [28,29,30]. The value of each sample was normalized with that of glyceraldehyde-3-phosphate dehydrogenase (GAPDH). The relative gene expression concentrations were evaluated using the 2^−ΔΔct^ method as described by Pfaffl [31].

### 2.12. Histopathology

Tissue sections from the liver of each animal were taken and processed for histopathological examination. The sections were directly fixed in 10% formalin solution, dehydrated in alcohols, cleared in xylene and embedded in paraffin blocks. Sections of 5 μm thickness were obtained and stained with hematoxylin and eosin [32].

### 2.13. Immunohistochemical Analysis of Glutathione S-Transferase P

Immunohistochemical staining (IHS) of glutathione S-transferase placental form (GST-P) was performed on all hepatic tissue samples using 4-μm thick paraffin-embedded sections. IHS for GST-P (MBL Co. Ltd., Nagoya, Japan) and analysis of GST-P-positive foci were performed as described previously by [33]. The percentage of cells with positive labeling of GST-P was determined by counting 1000 cells observed in 10 high-power fields (×400) in the pre-neoplastic foci and the surrounding hepatic tissue.

### 2.14. Statistical Analysis

Data were analyzed with One-way ANOVA and Tukey’s post hoc multiple range tests using GraphPad Prism 5 (GraphPad, San Diego, CA, USA). *p* value < 0.05 was considered statistically significant. All data are expressed as means± SD.

## 3. Results

### 3.1. UPLC-PDA-MS/MS for Metabolite Analysis

The analysis of the data in the negative mode ion revealed the presence of 40 compounds. The major identified constituents are citric acid, citric acid isomers, and octadecenoic, octadecadienoic, and octadecanoic acid derivatives. The complete chemical profile is shown in Figure 1 and listed in Table 2. This assortment of such bioactive compounds can explain the exhibited biological potential of *A. awamori* in our study. UPLC-MS data were processed using Shimadzu’s LabSolutions software. The compounds were identified based on the molecular weight determined from the most intense adduct ion found in the full MS spectrum of each compound, MS/MS fragmentation ions, or neutral losses found in full MS or MS/MS spectra and, whenever possible, maximum absorption wavelength from PDA spectra. MassBank and FooDB databases were used as references in identifying the eluted compounds in addition to the published data in the literature. Detailed description of all bioactive compounds is given in Appendix A.

### 3.2. Total Content of Flavonoids and Polyphenols

The total flavonoid content was measured as 6.78 mg/g equivalent to rutin and the total polyphenol content was measured as 7.36 mg/g equivalent to gallic acid.

### 3.3. Body and Liver Weights

In rats supplemented only by different concentrations of *A. awamori*, there was a non-significant effect on the body weight, liver weight, and liver weight/body ratio, except for a significant decrease in body weight in the group that received 1 mg/kg BW *A. awamori*. However, the DEN treated group exhibited a significant decrease (*p* < 0.001) in body weight when compared with the control untreated group (Table 3). While, a marked increase in liver weight and liver to body weight ratio was observed when compared with the control untreated rats. On the contrary, in the DEN injected groups supplemented with *A. awamori* (DEN + ASP1, DEN + ASP0.5, and DEN + ASP0.25), a significant improvement in the body weight, liver weight and liver to body weight ratio was noted. All effects on the body weight, liver weight, and liver weight/body ratio are shown in Table 3.

### 3.4. Hematological Examination

The erythrogram showed non-significant changes in RBC count, Hb, HCT, MCV, MCH and MCHC between the different experimental groups. However, the leucogram revealed that the DEN administration induced a significant increase of WBCs (*p* < 0.001), lymphocyte (*p* < 0.01), monocyte (*p* < 0.001) and neutrophil (*p* < 0.01) counts when compared with the control untreated animals. On the other hand, the groups supplemented with *A. awamori* in different doses along with DEN (DEN + ASP1, DEN + ASP0.5, DEN + ASP0.25) showed an improvement in the WBC, lymphocyte, monocyte and neutrophil counts compared with the group injected with DEN only as listed in Table 4.

### 3.5. Biochemical Findings

Non-significant changes were observed in rats supplemented only with different concentrations of *A. awamori*, while rats administered with DEN showed a significant decrease (*p* < 0.001) in the levels of serum total proteins, albumin and globulins (Figure 2I). Furthermore, a significant increase *(p* < 0.001) in the activities of ALT, AST, GGT and ALP (Figure 2I) was seen when compared with the untreated control animals. Conversely, in groups supplemented with *A. awamori* (DEN + ASP1, DEN + ASP0.5, DEN + ASP0.25), there was a significant increase in serum total proteins, albumin, and globulin levels when compared with the DEN administered group. Moreover, *A. awamori* supplementation significantly reduced the elevated enzyme activities of serum ALT, AST, GGT, and ALP, which were induced by DEN administration as shown in Figure 2I.

### 3.6. Hepatic Oxidative Stress and Antioxidant Status

Non-significant changes were observed in the levels of MDA, GSH and CAT activities in rats that only received different doses of *A. awamori*. However, in the DEN treated group, the hepatic MDA levels were significantly increased (*p* < 0.001), while GSH levels and catalase activities were significantly decreased (*p* < 0.001) in comparison with control untreated rats. Inversely, in *A. awamori* supplemented groups (DEN + ASP1, DEN + ASP0.5, DEN + ASP0.25), the MDA levels were significantly reduced with a significant improvement in GSH levels and CAT activities when compared with the DEN treated group. Levels of MDA, GSH and CAT are presented in Figure 2II.

### 3.7. Genes Expression Level of Cyp19 and p53

Gene expression revealed significant up-regulation of *Cyp19* mRNA (*p* < 0.001) in the DEN group as compared with the untreated control group. While, this was significantly down-regulated (*p* < 0.001) in rats given DEN and supplemented with *A. awamori* (DEN + ASP1, DEN + ASP0.5, DEN + ASP0.25) as compared with the DEN treated group. However, the expression level of *p53* mRNA was significantly down-regulated (*p* < 0.001) in the DEN group as compared with the control untreated group. In groups supplemented with *A. awamori* (DEN + ASP1, DEN + ASP0.5, DEN + ASP0.25), there was a significant increase in p53 expression level in comparison with the DEN treated group (Figure 2III).

### 3.8. Histopathology

Liver sections from the control untreated rats (Figure 3I-A) and rats treated only with different concentrations of *A. awamori* (Figure 3I-B) showed normal hepatocellular architecture comprising hepatocytes with typical cytoplasm and small regular nuclei radially arranged in a radial pattern around the central vein. In contrast, the animals treated with DEN showed loss of normal architecture with irregularly formed hepatocytes with increased nuclear to cytoplasmic ratio in addition to extensive steatosis cells with cytoplasmic vacuolation (Figure 3I-C). However, in rats supplemented with *A. awamori* and subjected to DEN (DEN + ASP1, DEN + ASP0.5, DEN + ASP0.25), there was a marked decrease in DEN’s hepatic lesions and reversal of liver architecture (Figure 3I-D–F).

### 3.9. Immunohistochemistry of GST-P Expression

Liver tissues from untreated control and *A. awamori* sole supplied groups had no positive cancer cells for GST-P. However, the immunohistochemical labeling of GST-P during the initiation of hepatocarcinogenesis induced by DEN revealed widely distributed GST-P-positive hepatocytes in a percentage of 43.33% (Figure 3II-A). Interestingly, GST-P-positive foci were significantly decreased (*p* ≤ 0.05) with *A. awamori* supplementation in a dose dependent manner, where it decreased to 16.33% in the DEN + ASP1 group, to 30.33% in the DEN + ASP0.5 group and to 36.6% in the DEN + ASP0.25 group (Figure 3II-B–D). The immunoreactivity of GST-P is summarized in Table 5 and Figure 3II.

## 4. Discussion

The current work was planned to elucidate the modulatory effect of *A. awomori* against the initiation of DEN induced hepatocarcinogenesis. DEN is a ROS-generating carcinogen that results in formation of preneoplastic foci [8,34,35]. These generated ROS are usually deactivated by the endogenous antioxidants [36]. Consequently, antioxidants act as the cellular housekeepers via mopping up of free radicals before inducing DNA damage [8].

The current study revealed that oral supplementation of *A. awamori* alone in different concentrations did not have a significant effect on the body weight, liver weight, and liver weight/body ratio. Furthermore, non-significant changes in the erythrogram, leukogram, serum protein profile and liver injury markers indicated that dietary supplementation of *A. awamori* is safe and useful with no evidence of cellular oxidative damage. In the same context, numerous studies stated the beneficial effect of *A. awamori* regarding its antioxidant potential [23,37,38].

The observed reduction in the rat’s body weight with significant increase of liver weight by DEN administration may be due to the enhancement in the metabolic activity of the body systems. The same findings were recorded in previous studies [39,40]. The increased hepatic weight may be due to cellular swelling (shift of extracellular water into the cells), which is the primary indicator for cell injury together with fatty changes of the liver [41].

Furthermore, our results demonstrated that DEN administration resulted in significant changes in the leucogram as compared with the control untreated group, which is in line with Holsapple et al. [42]. These changes in the leucogram were ameliorated with *A. awamori* supplementation. Furthermore, DEN/*A. awamori* supplementation restored the hepatic injury markers to normal with reduction of the MDA levels and a significant enhancement in antioxidant status. The increased level of MDA in HCC is an indicator for cell membrane injury and alterations in organ structure [43,44,45], which is clearly supported by our findings.

The antioxidant properties of *A. awamori* are related to its content of several bioactive compounds. Few data are available about the chemical principles of *A. awamori* [24]. For this reason, the current study aimed to investigate the metabolic fingerprint of this fungus. Citric acid was found to induce apoptosis in gastric carcinoma via the mitochondrial pathway [46]. Ren et al. [47] documented that citrate suppressed growth of different cell lines by inhibiting glycolysis. Moreover, citric acid, citric acid isomers and ascorbic acid are well recognized antioxidants, and reduced the hepatocellular injury induced by carbon tetrachloride in rats [48]. Additionally, Richard et al. [49] indicated that polyunsaturated fatty acids could act as antioxidants through their free radical scavenging ability.

In addition, the observed hepatoprotective effect of *A.awamori* in the DEN-induced HCC model may be attributed to its total flavonoid and total polyphenol contents. Polyphenols employ anti-proliferative and anti-angiogenic effects that are principally obvious in tumor cells [50,51].

Histopathological examination of the liver tissues from DEN treated rats showed marked alteration in the tissue architecture, indicating carcinogenicity, which is consistent with the results of Abdelhady et al. [8] who reported hypertrophy of centro-lobular hepatocytes, cytoplasmic vacuolation, and periductal oval cell proliferation with an increase in mitotic figures after one week from a single dose injection of DEN. In addition, Salah et al. [52] confirmed that the hepatic neoplastic changes induced by DEN are due to ROS generation. In rats post-treated with different concentrations of *A. awamori* after DEN administration, restoration of hepatic architecture with an obvious decrease in DEN’s hepatic lesions was observed. Overexpression of GST-P in response to DEN indicates the initiation of hepatic carcinogenesis. These GST-P-positive foci were significantly decreased (*p* ≤ 0.05) with *A. awamori* supplementation in a dose dependent manner. Abdo et al. [35] detected an increase in the number and area of GST-P-positive preneoplastic hepatic foci induced by DEN. Abdelhady et al. [8] also recorded marked GST-P immune-labeling within the neoplastic hepatocytes after one week in a group treated with a single dose of DEN. Xin you et al. [53] observed that the GST-P positive foci increased in all rat groups treated with an intra-peritoneal dose of DEN (200 mg/kg BW) and decreased in a dose dependent manner with resveratrol treatment.

*Cyp19* is a chief tumorigenesis enzyme that shares in chemotherapeutic activation in cancerous tissues [54]. *Cyp19* in rat liver carcinomas was significantly up-regulated, which is linked to invasion and is a probable HCC predictor [55]. In the present study, DEN significantly up-regulated *Cyp19*, indicating tumor initiation. Attenuation of *Cyp19* was observed in rats supplemented with *A. awamori.* These findings are firstly recording the effect of *A. awamori* on DEN cancer initiation through down-regulation of *Cyp19*.

The *p53* gene is the genome guardian and mainly regulates the proliferation, development, and transformation of the cells [56]. *P53* protein level is low in 50% of human tumors [57] and tends to be crucial for hepatocarcinogenesis [58]. Interestingly, genomic mutations in *p53* have been seen in several phases of malignancy [56]. In the present study, significant down-regulation (*P* < 0.001) of *p53* mRNA was observed in the DEN treated group, which is in line with [59,60]. Upon supplementation with *A. awamori*, a significant up-regulation of *p53* expression level was noticed. This may be due to *A. awamori* promoting de novo synthesis of *p53* protein or some other proteins to stabilize *p53*. To date, there are no previous reports about the effect of *A. awamori* on mRNA *expression of Cyp19* and *p53* in DEN-induced hepatocarcinogenesis. *A. awamori* achieved the equilibrium of cell proliferation and apoptosis through up-regulation of *p53* together with down-regulation of *Cyp19* to hinder tumor initiation and progression.

## 5. Conclusions

The current study proved for the first time that administration of *A. awomori* counteracts the adverse effects of DEN and initiated HCC in the Wistar rat model through restoring hepatic integrity by depletion of MDA level, thus lowering oxidative stress in addition to remarkable enhancement of hepatic antioxidant GSH level and CAT activities. Furthermore, reduction of GST-P positive foci, stabilization of some gene expression via down-regulating *Cyp19* and up regulation of *p53* gene occurred.

The presence of certain chemical constituents (citric acid, citric acid isomers, and octadecenoic, octadecadienoic and octadecanoic acid’s derivatives), polyphenols and flavonoids can explain the biological activity of *A. awamori*, which may be considered as antioxidant and anticancer agents. Based on these findings, *A. awamori* could be used as a natural hepatoprotective agent against chemical induced carcinogenicity. However, detailed further studies are needed to assess the effect of *A. awamori* in a long duration study with a wide profiling of gene expression levels in HCC induced by DEN.

## Figures and Tables

**Figure 1 antioxidants-10-00922-f001:**
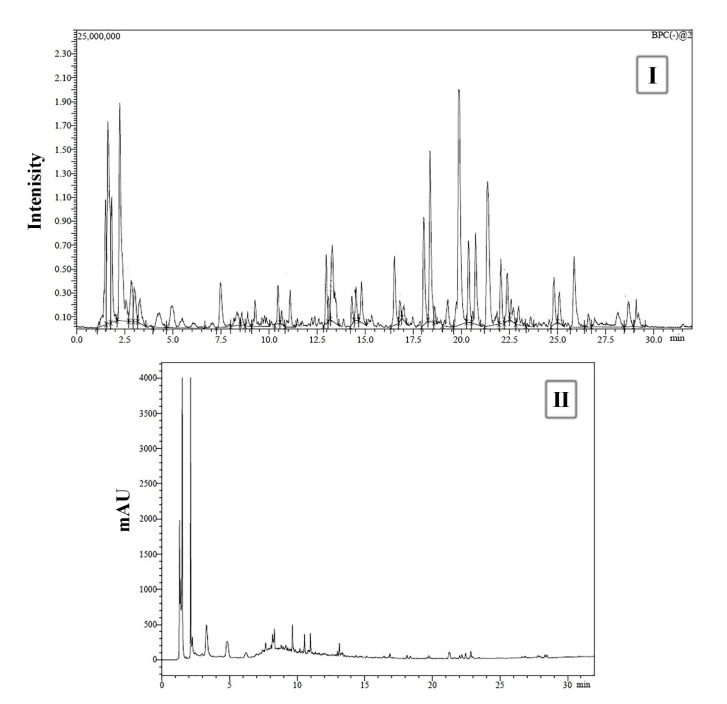
(**I**) UPLC-ESI-MS chromatogram of *A. awamori* aqueous ethanol extract in negative ionization mode. (**II**) UPLC-PDA chromatogram of *A. awamori* aqueous ethanol extract.

**Figure 2 antioxidants-10-00922-f002:**
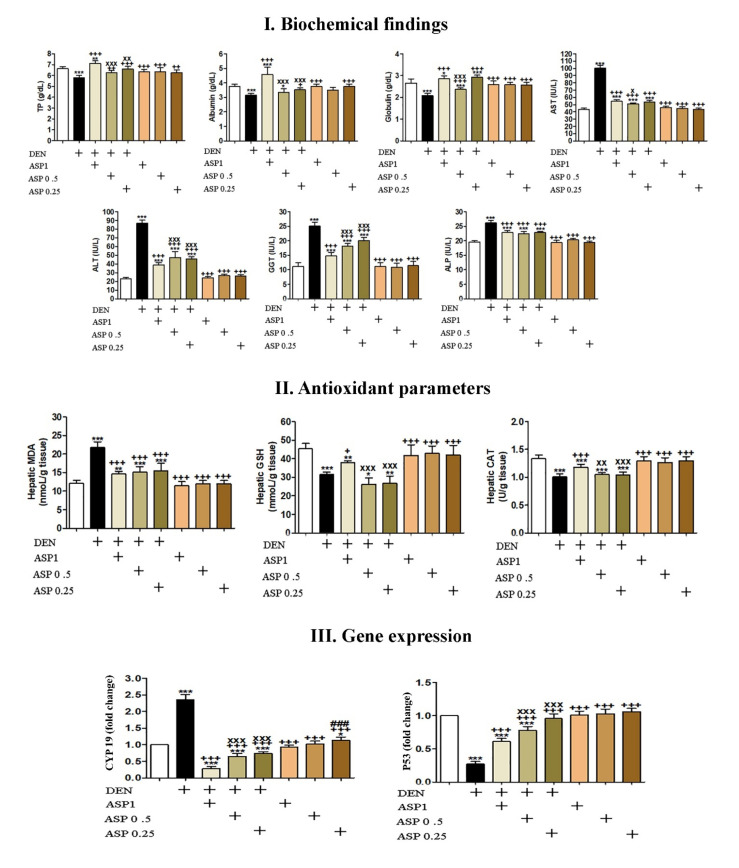
(**I**) Biochemical assessment of: Total proteins (TP), albumin, globulins, AST, ALT, GGT, and ALP. (**II**) Hepatic oxidative stress and antioxidant status: MDA, GSH, CAT. Data were analyzed with one-way ANOVA followed by Tukey’s multiple comparison test. * *p* < 0.05, ** *p* < 0.01, and *** *p* < 0.001 vs. Control. ^+^
*p* < 0.05, ^++^
*p* < 0.01 and ^+++^
*p* < 0.001 vs. DEN. ^x^
*p* < 0.05, ^xx^
*p* < 0.01, and ^xxx^
*p* < 0.001 vs. DEN + ASP1. Error bars represent mean ± SD. *n* = 9. (**III**) Expression fold changes of cytochrome P450 (*Cyp19*) and p53 genes in the liver. Data were analyzed with one-way ANOVA followed by Tukey’s multiple comparison test. * *p* < 0.05 and *** *p* < 0.001 vs. Control. ^+++^
*p* < 0.001 vs. DEN. ^xxx^
*p* < 0.001 vs. DEN + ASP1. ^###^
*p* < 0.001 vs. ASP1. Error bars represent mean ± SD. *n* = 9. Where: DEN: DEN supplied group, ASP1: group received *A. awamori* 1 mg/kg b.wt, ASP0.5: group received *A. awamori* 0.5 mg/kg b.wt, ASP0.25: group received *A. awamori* 0.25 mg/kg b.wt.

**Figure 3 antioxidants-10-00922-f003:**
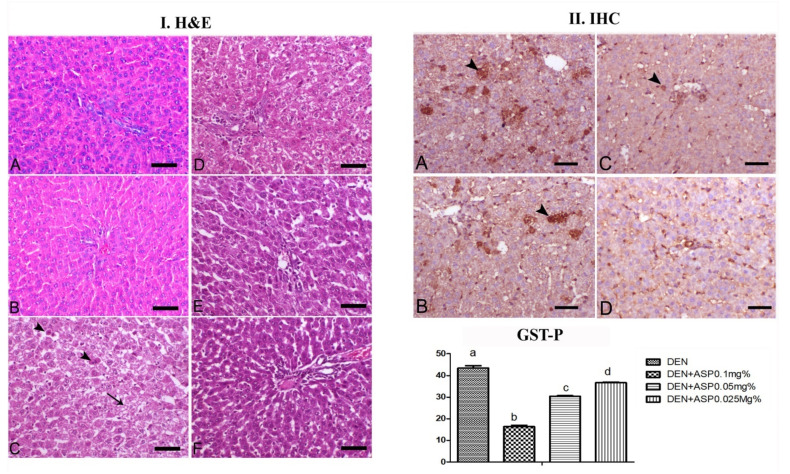
(**I**) Histopathological examinations of the liver in different experimental groups (**A**,**B**): Control untreated group showed normal liver structures; (**C**): DEN treated groups showed hepatocytes vacuolation (arrow) with an increase in mitosis (arrowheads). (**D**–**F**)**:** DEN + ASP1, DEN + ASP0.5, and DEN + ASP0.25, respectively, showed a moderate degree of hepatocyte vacuolation with a marked decrease in mitosis. H&E, bar= 50 µm. (**II**) Immunolabelling of glutathione S-transferase placental form (GST-P) in livers of rats in different experimental groups. (**A**): DEN group, (**B**) DEN/ASP1 group, (**C**), DEN/ASP0.5 group, (**D**), DEN/ASP0.25 group. Arrowheads point to GST-P positive foci (brown color). Scale bar= 50 μm. Values with different letters differ significantly at (*p* ≤ 0.05).

**Table 1 antioxidants-10-00922-t001:** Primers’ sequences.

Genes	5′→3′	References
*CYP19*	F: CGTCATGTTGCTTCTCATCG	[28]
	R: TACCGCAGGCTCTCGTTAAT	
*p53*	F: CCCAGGGAGTGCAAAGAGAG	[29]
	R: TCTCGGAACATCTCGAAGCG	
*GAPDH* *	F: TGCACCACCAACTGCTTAGC	[30]
	R: GGCATGGACTGTGGTCATGAG	

* housekeeping gene.

**Table 2 antioxidants-10-00922-t002:** UPLC-PDA-MS/MS profiling data for *A. awamori*.

NO.	R_t_ Min	[M-H]^-^ *m*/*z*	MS/MS Ions *m*/*z*	Identification
1	1.51	195	177, 159, 129, 99, 75	Gluconic acid
2	1.63	191	129, 111, 87, 85	Citric acid isomer 1
3	1.82	191	129, 111, 87, 85	Citric acid isomer 2
4	2.25	191	129, 111, 87, 85	Citric acid
5	2.84	205	143, 111, 87	Methyl derivative of citric acid
6	2.99	147	147, 103	Cinnamic acid
7	3.30	161	129, 101, 85, 83	Methyl cinnamate
8	4.31	219	187, 159, 157, 143, 129, 125, 115,111, 97, 87	1,3-Dimethyl citrate
9	4.96	153	109	Gentisic acid
10	6.52	219	157, 113, 111, 87	1,5-Dimethyl Citrate isomer
11	7.48	175	157, 129, 115, 113, 85	L-ascorbic acid
12	8.34	563	545, 503, 473, 443, 425, 412, 406, 383, 365, 353, 311, 378, 293, 233	Apigenin-6,8-diglucoside (vicenin II)
13	8.59	165	121	Phthalic acid
14	8.89	415	191, 175, 139, 119, 101, 89	Tetrahydroxy coumarin-3′-carboxylic acid-β-D-glucoside
15	9.27	151	107	Anisic acid
16	10.12	163	119	*p*-Coumaric acid
17	10.66	187	169, 143, 125, 123, 97	Benzoic acid or gallic acid derivative
18	11.10	221	206, 162, 150, 133	Isofraxidin
19	11.12	439	289, 274, 247, 175, 149, 134	Unknown
20	12.98	331	313, 295, 201, 171, 157, 127	Trihydroxy octadecanoic acid
21	13.09	663	447, 234,215, 203, 157, 125, 124, 111	Unknown
22	13.29	329	314, 299, 271, 229, 211, 171, 157, 139, 127, 99	Trihydroxy octadecenoic acid
23	13.50	413	252, 234, 193, 175, 163, 134, 119	Methyl-dihydroxy-dihydrocoumarin-3′-carboxymethyl-β-D-glucoside
24	14.31	1085	865	Derivative of procyanidin C1
25	a-14.53 b-14.81	329	293, 275, 211, 201, 181, 171, 155, 139, 127	Trihydroxy octadecenoic acid isomers
26	16.53	499	467, 455, 439, 423	Ursolic acid derivative
27	16.80	287	287, 269, 251, 225, 201, 155	Aromadendrin
28	17.01	311	293, 275, 255, 223, 183	15,16-Dihydroxy-9,12-octadecadienoic acid
29	a-18.08 b-18.37	313	277, 201, 171, 165, 155, 127	10-Hydroperoxy-8E-octadecenoic acid and its isomer
30	18.61	485	485, 439, 391	Methoxy ursolic acid
31	a-19.29 b-19.87	315	313, 297, 279, 201, 171, 155, 141, 127	Dihydroxy-octadecanoic acid and its siomer
32	a-20.38 b-20.74	355	313, 295, 277, 201, 171, 123	Propyl ester of compound 313 and its isomer
33	21.37	295	277, 195, 171, 113	13(S)-hydroxyoctadecadienoic acid (α-Artemisolic acid)
34	25.86	295	249, 193, 155, 141	Isomer of compound 28
35	a-22.05 b-22.39	357	315, 297, 279, 171	Propyl ester of compound 26 and its isomer
36	24.82	333	279, 265, 155, 139, 127	Unknown
37	26.59	509	491, 465, 447, 421, 359, 347, 325, 295, 181	Unknown
38	28.68	297	251	6,7-epoxystearic acid
39	28.15	271	253, 225	2-Hydroxy palmitic acid
40	29.09	392	130	Unknown

**Table 3 antioxidants-10-00922-t003:** Body weight, liver weight and relative weight changes of control, DEN and *Aspergillus awameri* supplemented rat groups.

Groups	Items					
	Initial Weight (kg/Animal)	Final Weight (kg/Animal)	Body Gain (kg/Animal)	Gain Relative Weight *	Liver Weight	Liver Relative Weight #
Control	0.164 ± 0.01 ^a^	0.227 ± 0.05 ^a^	0.063 ± 0.04 ^a^	27.75 ± 18.19 ^a^	5.98 ± 0.20 ^b^	2.63 ^b^
DEN	0.164 ± 0.02 ^a^	0.144 ± 0.02 ^d^	-0.02 ± 0.01 ^d^	-13.89 ± 7.21 ^d^	7.28 ± 0.54 ^a^	5.06 ^a^
DEN + ASP (0.1 mg/kg BW)	0.163 ± 0.02 ^a^	0.197 ± 0.01 ^b^	0.034 ± 0.01 ^b^	17.26 ± 11.49 ^b^	6.02 ± 0.33 ^b^	3.06 ^b^
DEN + ASP(0.05 mg/kgBW)	0.166 ± 0.02 ^a^	0.198 ± 0.02 ^b^	0.032 ± 0.01 ^b^	16.16 ± 7.07 ^b^	5.75 ± 0.45 ^b^	2.90 ^b^
DEN + ASP(0.025 mg/kgB)	0.167 ± 0.02 ^a^	0.172 ± 0.03 ^c^	0.005 ± 0.02 ^c^	2.91 ± 1.01 ^c^	5.84 ± 0.37 ^b^	3.34 ^b^
ASP(0.1 mg/kg BW)	0.164 ± 0.02 ^a^	0.187 ± 0.02 ^b^	0.023 ± 0.02 ^b^	12.3 ± 4.11 ^b^	5.64 ± 0.38 ^b^	3.02 ^b^
ASP(0.05 mg/kg BW)	0.161 ± 0.02 ^a^	0.219 ± 0.02 ^a^	0.058 ± 0.01 ^a^	26.48 ± 8.61 ^a^	5.91 ± 0.37 ^b^	2.70 ^b^
ASP(0.025 mg/kg BW)	0.161 ± 0.03 ^a^	0.222 ± 0.06 ^a^	0.061 ± 0.04 ^a^	27.48 ± 21.12 ^a^	5.68 ± 0.22 ^b^	2.56 ^b^

* Body gain relative to the initial weight. Values are means ± standard error. Mean values with different letters at the same column differ significantly at (*p* ≤ 0.05). # Liver relative weight = (liver weight/Final weight). Values are means ± standard error. Mean values with different letters at the same column differ significantly at (*p* ≤ 0.05).

**Table 4 antioxidants-10-00922-t004:** Hematological findings of control, DEN and *Aspergillus awameri* supplemented rat groups.

Groups	Items									
	RBCS (10^6^/µL)	HGB (g/dL)	HCT (%)	MCV (fL)	MCH (pg)	MCHC (%)	WBCS (10^3^/µL)	Lymphocyte (10^3^/µL)	Neutrophils (10^3^/µL)	Monocytes (10^3^/µL)
Control	6.08 ± 0.08 ^a^	13.9 ± 0.10 ^a^	34.4 ± 0.18 ^a^	56.58 ± 0.92 ^a^	22.86 ± 0.20 ^a^	40.41 ± 0.41 ^a^	17.5 ± 0.32 ^b^	12.92 ± 0.10 ^b^	1.95 ± 0.23 ^b^	2.63 ± 0.12 ^b^
DEN	6.19 ± 0.19 ^a^	13.4 ± 0.49 ^a^	33.08 ± 1.2 ^a^	53.44 ± 1.10^a^	21.65 ± 0.41^a^	40.51 ± 0.19 ^a^	23.7 ± 0.20^a^	16.1 ± 0.34 ^a^	3.2 ± 0.18 ^a^	4.4 ± 0.15 ^a^
DEN + ASP (0.1 mg/kg BW)	5.66 ± 0.11 ^a^	12.4 ± 0.37 ^a^	31.5 ± 0.98 ^a^	55.65 ± 0.97 ^a^	21.9 ± 0.34 ^a^	39.37 ± 0.18 ^a^	17.75 ± 0.7 ^b^	12.9 ± 0.56 ^b^	2.2 ± 0.30 ^b^	2.65 ± 0.145 ^b^
DEN + ASP (0.05 mg/kg BW)	5.93 ± 0.18 ^a^	12.8 ± 0.36 ^a^	32.1 ± 0.96 ^a^	54.13 ± 0.84 ^a^	21.6 ± 0.27 ^a^	39.88 ± 0.11 ^a^	17.6 ± 0.76 ^b^	11.77 ± 0.5 ^b^	2.7 ± 0.20 ^b^	3.13 ± 0.313 ^b^
DEN + ASP (0.025 mg/kg BW)	6.15 ± 0.12 ^a^	13.2 ± 0.38 ^a^	33.5 ± 1.04 ^a^	54.47 ± 0.84 ^a^	21.5 ± 0.34 ^a^	39.40 ±0.17 ^a^	18.2 ± 0. 67 ^b^	11.8 ± 0.37 ^b^	2.93 ± 0.21 ^a^	3.44 ± 0.25 ^b^
ASP (0.1 mg/kg BW)	5.72 ± 0.10 ^a^	12.8 ± 0.31 ^a^	32.7 ± 0.85 ^a^	57.17 ± 0.65 ^a^	22.38 ± 0.29 ^a^	40.10 ± 0.177 ^a^	17.4 ± 0.45 ^b^	12.9 ± 0.27 ^b^	1.93 ± 0.19 ^b^	2.57 ± 0.38 ^b^
ASP(0.05 mg/kg BW)	5.90 ± 0.19 ^a^	12.6 ± 0.11 ^a^	31.7 ± 0.40 ^a^	53.73 ± 0.60 ^a^	21.36 ± 0.34 ^a^	39.4 ± 0.23 ^a^	16.13 ± 0.06 ^b^	11.5 ± 0.57 ^b^	1.83 ± 0.31 ^b^	2.8 ± 0.26 ^b^
ASP(0.025 mg/kg BW)	5.75 ± 0.09 ^a^	12.6 ± 0.38 ^a^	32.2 ± 1.08 ^a^	55.7 ± 0.8 ^a^	21.9 ± 0.29 ^a^	39.13 ± 0.16 ^a^	17.3 ± 0.44 ^b^	12.7 ± 0.56 ^b^	2.06 ± 0.38 ^b^	2.53 ± 0.29 ^b^

Mean values with different letters at the same column differ significantly at (*p* ≤ 0.05).

**Table 5 antioxidants-10-00922-t005:** Scoring summary of GST-P immunolabelling in different experimental groups in the liver tissues in a rat model.

Groups	IHC
Control	Zero
DEN	43.33 ± 1.19 ^a^
DEN + ASP (0.1 mg/kg BW)	16.33 ± 0.6 ^b^
DEN + ASP (0.05 mg/kg BW)	30.33 ± 0.37 ^c^
DEN + ASP (0.025 mg/kg BW)	36.6 ± 0.33 ^d^
ASP (0.1 mg/kg BW)	Zero
ASP (0.05 mg/kg BW)	Zero
ASP (0.025 mg/kg BW)	Zero

Mean values with different letters at the same column differ significantly at (*p* ≤ 0.05).

## Data Availability

The authors confirm that the data supporting the findings of this study are available within the article [and/or] its Appendix A.

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
