# Peer review of "Ameliorative Effects of Aspergillus awamori against the Initiation of Hepatocarcinogenesis Induced by Diethylnitrosamine in a Rat Model: Regulation of Cyp19 and p53 Gene Expression"

_antioxidants, 2021, doi:10.3390/antiox10060922_

Round 1

Reviewer 1 Report

The authors have addressed most of my original comments. However, there is still one question that needs to be addressed and some changes to the discussion are required.

Since GSH levels were decreased in DEN treated rats, what would the effects of NAC supplementation be?

Page 16, 2 nd paragraph, starting with "It was suggested......" needs to be removed, it has nothing to do with the present study and does not add to the discussion.

Author Response

Thank you for your time and valuable comments concerning our manuscript. The authors have studied the comments which were very helpful to allow us to add more details to our work, also we have made corrections according to the reviewers’ comments.

1- Since GSH levels were decreased in DEN treated rats, what would the effects of NAC supplementation be?

It is not clear what is meant by NAC. If it means N-acetyl-cysteine  (NAC), previous studies (El-Shorbagy et al., 2019) mentioned that NAC manifests a promising effect in cancer prevention as they possess anti-carcinogenic properties. In vitro NAC has been reported to inhibit the growth and invasive behavior of human cancer cells, such as colorectal, bladder, prostate, tongue and lung carcinoma cell. Which we expect if used in the same experiment will have a positive effect on the prevention of cancer. However, this point needs to be examined.

2- Page 16, 2nd paragraph, starting with "It was suggested......" needs to be removed, it has nothing to do with the present study and does not add to the discussion.

This paragraph has been removed from the discussion

Reviewer 2 Report

This manuscript contains extensive amount of in vivo data which support the protective effect of Aspergillus awamori against the initiation of hepatocellular carcinoma in rats. However, how the administration of Aspergillus awamori affects the survival of the animals was not investigated. Moreover, the manuscript contains several grammatical errors; the authors should check for spelling of words that are distributed throughout the manuscript. In addition, the quality of the figures is low and their legends do not provide sufficient information to understand the presented data. The detailed description of the identification of the compounds (pages 8-12) should be removed from the main text of the manuscript and presented as a supplement. The discussion, albeit very interesting, appears largely speculative as several aspects have not been tested. Therefore, I do not support the current version of this manuscript for publication in this Journal.

Author Response

Thank you for your time and valuable comments concerning our manuscript. The authors have studied the comments which were very helpful to allow us to add more details to our work, also we have made corrections according to the reviewers’ comments.

Reviewer2

1- This manuscript contains extensive amount of in vivo data which support the protective effect of Aspergillus awamori against the initiation of hepatocellular carcinoma in rats. However, how the administration of Aspergillus awamori affects the survival of the animals was not investigated.

The used dose of DEN was appropriate for induction of carcinogenesis not toxic dose to cause mortality. All rats used in the experiment survived till the end of the experiment without mortality.

2- Moreover, the manuscript contains several grammatical errors; the authors should check for spelling of words that are distributed throughout the manuscript.

The spelling have been checked along the whole manuscript

3- In addition, the quality of the figures is low and their legends do not provide sufficient information to understand the presented data.

All figures have been modified for quality and the legends have been checked.

4- The detailed description of the identification of the compounds (pages 8-12) should be removed from the main text of the manuscript and presented as a supplement.

This part has been removed and presented in the supplementary data file under ''supplementary file 2''

5- The discussion, albeit very interesting, appears largely speculative as several aspects have not been tested.

The discussion part has been modified and all the aspect discussed are related to the results of the recent work.

Round 2

Reviewer 2 Report

The Manuscript was substantially improved. Therefore, I suggest it for publication with a few minor corrections and checking the grammar throughout the text. The resolution of Figure 1 is still low. The result displayed on a bar graph in Figure 2 lacks the labeling of the y axis and is not explained in the corresponding legend.

Author Response

thank you for your time and valuable comments

Responses to the reviewers' comments include:

1- The resolution of Figure 1 is still low.

The resolution of figure 1 has been modified

2- The result displayed on a bar graph in Figure 2 lacks the labeling of the y axis and is not explained in the corresponding legend.

Figure 2 has been corrected

3- checking the grammar throughout the text

Grammar has been checked 

This manuscript is a resubmission of an earlier submission. The following is a list of the peer review reports and author responses from that submission.

Round 1

Reviewer 1 Report

This paper is an excellent experimental investigation of hepatocellular carcinoma. It can be published in its current format although I consider it excessively long. Perhaps groups 4, 5, 7 and 8 can be excluded from text respectively. Finally, the Discussion should explain how we can transfer the data obtained from this research work to the human clinic.

Reviewer 2 Report

The aim of this study is to demonstrate that DEN induces the initiation of hepatocarcinogenesis can be suppressed by A. awamori and the authors try to explain which compounds are involved in the inhibition.

  1. The DEN induced hepatocarcinogenesis in rat models:

Rats are induced liver cancer by DEN treatment at least for 10-20 weeks, however, the authors only observed for 8 days. The reviewer understands that the author would like to emphasize the “initiation” of hepatocarcinogenesis, but DEN treatment for 8 days may just trigger immunologic derangement not even induce hepatitis. Therefore, it is questionable whether initiation of hepatocarcinogenesis is induced in this study.

  1. Too much explanation of results in the discussion section. The authors should reorganize the result and discussion section.

Reviewer 3 Report

Dietary consumption of Aspergillus awamori (SA) a fungus from Japan has been shown to have health benefits. These have been attributed to its constituents comprising of coumarins, saponins, flavonoids, and tannin, which have antioxidant properties. In the present study, the authors evaluated the effects of SA in diethylnitrosamine (DEN) induced liver cancer in rats.

Comments:

  1. The protocol described is a prevention protocol and not a treatment one. Therefore, all statements, such as the last line of abstract, need to be changed.
  2. It is not clear why the authors chose to evaluate the effects of SA on CYP19 (aromatase). A good number of procarcinogens are converted to carcinogens by CYP1A1 and/or CYP1A2.
  3. Metabolite analysis is quite robust. Which constituent(s) contributes to the overall activity of SA?
  4. Since GSH levels were decreased in DEN treated rats, what would the effects of NAC supplementation be?
  5. The Discussion is far too long and repeats the results.
  6. A good amount of the Discussion has nothing to do with the current study.